# Understanding the Mechanisms of Diet and Outcomes in Colon, Prostate, and Breast Cancer; Malignant Gliomas; and Cancer Patients on Immunotherapy

**DOI:** 10.3390/nu12082226

**Published:** 2020-07-26

**Authors:** Shivtaj Mann, Manreet Sidhu, Krisstina Gowin

**Affiliations:** 1Division of Hematology and Oncology, University of Arizona, Tucson, AZ 85719, USA; tajmann@arizona.edu; 2Division of Family Medicine, Rowan University, Stratford, NJ 08084, USA; sidhumk@rowan.edu

**Keywords:** diet, cancer, carcinoma, colon cancer, colorectal cancer, prostate cancer, breast cancer, malignant gliomas, immunotherapy, cancer outcomes, cancer prognosis, western diet

## Abstract

Cancer patients often ask which foods would be best to consume to improve outcomes. This is a difficult question to answer as there are no case-controlled, prospective studies that control for confounding factors. Therefore, a literature review utilizing PubMed was conducted with the goal to find evidence-based support for certain diets in specific cancer patients—specifically, we reviewed data for colon cancer, prostate cancer, breast cancer, malignant gliomas, and cancer patients on immunotherapy. Improved outcomes in colon cancer and patients on immunotherapy were found with high-fiber diets. Improved outcomes in malignant gliomas were found with ketogenic diets. Improved outcomes in prostate cancer and breast cancer were found with plant-based diets. However, the data are not conclusive for breast cancer. Additionally, the increased intake of omega-3 fatty acids were also associated with better outcomes for prostate cancer. While current research, especially in humans, is minimal, the studies discussed in this review provide the groundwork for future research to further investigate the role of dietary intervention in improving cancer outcomes.

## 1. Introduction

A commonly encountered question posed by cancer patients is understandably in the form of “What should I eat?”. This is a rather difficult question to answer as there are no case-controlled, prospective studies that examine the effects of specific diets on cancer outcomes, while controlling for confounding factors such as obesity, tobacco use, and exercise. The demand is high from cancer survivors and physicians to provide evidence-based diet recommendations, but the data are limited. While it is true we have learned that a diet rich in processed foods, simple sugars, and processed meats is carcinogenic, the current recommendations appear superficial, lacking a deeper understanding of underlying mechanisms and thought processes. This has created an overwhelming need to better understand the physiological effect of dietary intervention, and how it can be used to promote the efficacy of cancer therapy, enhance immune function, and ultimately improve prognosis.

Most of the data examining the link between diet and cancer outcomes originate from in vitro and in vivo animal studies, that are then extrapolated and applied to humans. These data are muddied by inconsistent models, inconsistent terminologies, and inconsistent standards. For instance, common terminology used in the literature is “western diet;” however, there is no consensus on its meaning. Does it mean high in fat? High in meat? Highly processed? Low in fiber? A lack of fruits or vegetables? A combination of these? In reality, the human diet is far more complex, with innumerable socioeconomic factors that determine what a person consumes.

For example, the south Asian subcontinent is home to Dharmic religions. Its constituents mostly abide by a vegetarian diet; however, they are the largest consumers of milk in the world, often ingested in the whole-fat state [1]. In the literature, the abundant intake of animal products and fats, as is the case with the populace of south Asia who ingest large quantities of milk and cheese, is often labeled as a “western diet” and seen as a risk factor for oncogenesis. However, cancer rates for colorectal, breast, and prostate cancers are lower in south Asia than the United States [2]. This discordance between the dogmatic belief that the intake of animal products is oncogenic and the low rate of cancer in south Asia—the region with the highest intake of milk—shows us that oncogenesis, and likely cancer outcomes, are incredibly complex and influenced by host immunity, microbial exposure, and environmental exposure in addition to diet [3]. 

The following discussion attempts to reconcile the current research data available on diets that can modify outcomes in colon cancer, prostate cancer, malignant gliomas, breast cancer, and in cancer patients on immunotherapy. These cancers in particular were chosen because they comprise the majority of cancer cases seen, and they have the best preclinical and clinical data available regarding the role of diet in altering their outcomes. Such data in malignant hematology (lymphoma, leukemia, multiple myeloma, etc.) are incredibly sparse; as such, they were not included in the following discussion. We took a deeper look at the study designs and their implications, setting a basis for future studies on the role of specific diets in specific types of cancer. Furthermore, it is our aim to provide patients diagnosed with cancer and undergoing treatment with evidence-based data to help modify dietary behavior to improve outcomes.

## 2. Methods

A computer-aided literature search was conducted utilizing PubMed. The following keywords were used: “western diet,” “Mediterranean diet,” “diet and cancer,” “diet and immunotherapy,” “diet and cancer outcomes,” “diet and cancer prognosis,” “diet and central nervous system (CNS) malignancy,” “ketogenic diet and cancer,” “colon cancer and diet,” “colorectal carcinoma and diet,” “prostate cancer and diet,” “breast cancer and diet,” and “fiber and cancer.” Studies published in languages other than English were excluded. There were no exclusion criteria based on the date of publication. We included strong case series, in vitro and in vivo animal models, and human clinical data. Though a preference was given to studies focusing on human clinical subjects, such literature is quite limited. Therefore, we also relied on preclinical in vitro and animal model studies. In drawing connections and conclusions, the data from seven papers were selected for colorectal cancer, nine for prostate cancer, four for malignant gliomas, seven for breast cancer, and three for immunotherapy patients, as summarized in Table 1.

## 3. Colorectal Cancer

Colorectal carcinoma (CRC) is a worldwide problem. In terms of cancer statistics, it has the third highest incidence worldwide and the second highest mortality rate in the United States [31]. CRC is often labeled as a western, New World disease, as its incidence in nonurban areas, where the intake of processed foods and meats is rare, is almost nonexistent [32]. In fact, a prominent study by Van Blarigan et al. followed colon cancer patients in a prospective cohort study CALGB 89803. This study found that patients who abided by a diet rich in fruits, vegetables, and whole grains, had a 9% absolute risk reduction in death at the five-year point [7]. While this idea of a plant-based diet is widely prescribed, its mechanism of action is rarely analyzed or explained. In this section, we will look at both pre-clinical and clinical studies and try to examine the underlying mechanism of dietary intervention and how it improves clinical outcomes in CRC.

CRC, like other cancers, is believed to be related to locoregional inflammation. Though inflammation alone cannot initiate oncogenesis, it is believed to be a key participatory factor. Recently, there have been data to suggest that diet, gut microbiota, and gut environment can modulate gut inflammation and CRC oncogenesis, and may be modifiable variables in changing CRC outcomes [33].

To better understand the role of diet and lifestyle in modifying CRC risk, it is worthwhile to look at the basic biology of colonocytes. Colonocytes, unlike the other cells of our body, utilize butyrate and other short chain fatty acids (SCFAs), rather than glucose, as the primary source of energy [34]. Butyrate is not obtained directly from the diet; rather, it is a byproduct of fiber fermentation by gut microbiota. When fiber and other indigestible starches are ingested and pass through the small intestine and into the large colon, the local biota ferment these contents to produce SCFAs. These SCFAs in turn have physiological, anti-neoplastic, and anti-inflammatory activity.

Soret et al. demonstrated that a resistant starch diet in a rat model increased the expression of choline acetyltransferase in tandem with butyrate production and improved colon motility, thus highlighting the role of SCFAs like butyrate in maintaining colonic function [8]. In Crohn’s disease (CD), Segain et al. showed that human participants with CD who received butyrate enemas had improvement in disease as measured by post enema colonic biopsies. Moreover, butyrate enemas also decreased the levels of pro-inflammatory cytokines IL-1, IL-6, and TNF in peripheral blood, which was shown to be mediated by the butyrate-induced attenuation of NF-κB transcription activity [5]. These studies suggest that butyrate can regulate both the basic colon physiology and modulate locoregional inflammation. In addition to these findings, there is increasing interest in the relationship between gut microbiota and butyrate levels and their relation to CRC. The systematic review of literature pertaining to both patients and rat models has shown that in the presence of CRC, certain bacterial species (*Fusobacteria, Alistipes, Porphyromonadaceae, Coriobacteridae, Staphylococcaceae, Akkermansia spp. and Methanobacteriales*) are noted to be predominant and other species (*Bifidobacterium, Lactobacillus, Ruminococcus, Faecalibacterium spp., Roseburia, and Treponema*) are noted to be reduced. Along with this bacterial dysbiosis, colonic butyrate and SCFA levels are also noted to be diminished, which suggests that microbiota dysbiosis and diminished SCFA levels lead to locoregional inflammation. Taken together, it is postulated that certain gut flora could be pro-oncogenic, as it leads to diminished SCFA production, which has been shown to be pro-inflammatory in colonic tissue [6].

Additional studies have shown that certain gut flora could improve outcomes in both CRC cell models and patients. In vitro studies have shown that the presence of *Lactobacillus casei* and *Lactobacillis rhamnosus* supernatants prevents the invasion and progression of CRC. Though the actual mechanism is not entirely clear, it is known that *L. casei* and *L. rhamnosus* supernatants decrease levels of matrix metalloproteinase-9 (MMP-9) activity and the increase levels of the tight junction protein zona occludens-1 (ZO-1). Both proteins are known to regulate the extracellular matrix (ECM) of colonocytes, and MMP-9 is believed to be utilized by cancerous colonocytes to degrade the ECM in order to metastasize to local healthy tissue. In fact, in patients with CRC, MMP-9 is positively correlated with nodal metastasis [10].

Furthermore, in in vivo adenomatous polyposis coli (APC) gene mutated mice model studies, *Lactobacillus plantarum* has been noted to slow the progression of CRC. APC is a housekeeping gene and over 80% of CRC cases are noted to have a mutation in this gene. Yue et al. showed that APC-mutated mice given oral enemas of *L. plantarum* were noted to have a slower progression of CRC. Treatment with such enemas decreased the circulating levels of IL-6, IL-17, and TNF. In addition, enema treatment increased gut microbial diversity and increased the presence of *Firmicutes* and *Actinobacteria*, which mimics the gut flora of wild type mice. In the non-treated APC mice model, mice were noted to have decreased microbial diversity and higher levels of *Proteobacteria* and *Bacteroides* [9]. Again, no one single mechanism can explain how gut microbiota can modulate CRC outcomes. However, at least in the Yue et al. study, *L. plantarum* enema was shown to decrease inflammatory cytokines and beta-catenin protein expression in colonocyte nuclei. Beta-catenin levels modulate the Wnt signaling pathway, a pathway that is a key regulator in oncogenesis in APC-mutated CRC [35].

A recent publication by O’Keefe et al. evaluated human subjects and examined the role of diet in changing colonic inflammatory markers and SCFA levels. Twenty African Americans and 20 Africans each served as their own control and were recruited in the study to essentially swap their respective cultural diets. The 20 Americans were prescribed a diet that increased their average fiber intake from 14 g/d to 55 g/d and reduced their fat intake from 35% of calories to 16%. Africans were prescribed a diet that decreased their average fiber intake from 66 g/d to 12 g/d and increased their total calories from fat intake from 16% to 52%. Of great importance, the meals were prepared and administered under observation. The patients were monitored for two weeks, at which point colonic biopsies and stool studies were performed to determine the changes in inflammation and microbiota. After the 14-day period, colonic biopsies showed that the African Americans had a significantly lower epithelial proliferation rate compared to baseline, as measured by Ki67 staining. Colonic biopsies also showed a decreased presence of macrophages and lymphocytes, which are used as a marker for colonic inflammation. Lastly, abiding by a diet rich in fiber shifted the microbiota of the African Americans from one rich in *Bacteroides* to a more diverse flora, which included an increased presence of *Firmicutes, Roseburia,* and *Clostridium* species, which are known to be butyrate-producing bacteria. In fact, the high-fiber diet increased butyrate levels by 2.5 times in the African Americans. The western diet cut butyrate levels in the Africans and increased Ki67 staining in colonic biopsies, suggestive of increased inflammation [4].

Taken together, the above highlighted studies show that the gut microbiota plays a crucial role in regulating SCFA levels and that increasing dietary fiber can lead to the establishment of such favorable microbiota. Moreover, epidemiological studies do show that CRC is associated with a predominance of certain colonic microbial species and that such dysbiosis occurs concurrently with decreased SCFAs. In addition, in vitro studies show that byproducts of certain bacteria prevent CRC metastases by regulating MMP-9 and ZO-1. These findings suggest that increasing fiber intake could result in the diversification of the gut microbiome, the increased production of SCFAs, and as a result, improved CRC outcomes by regulating key ECM proteins. However, further investigation is needed, especially human clinical trials to further elucidate the effects of SCFAs on CRC.

## 4. Prostate Cancer

Prostate cancer is the most diagnosed cancer in men after skin cancer and has the second highest cancer-related mortality [36]. Historically, it was widely believed that fat intake was both a risk factor for the development of prostate cancer and likely a contributor for worse outcomes [37]. In fact, animal model studies have provided proof for this claim; Ngo et al., among others, showed that immunodeficient mice, engrafted with a prostate cancer xenograft, had a slower tumor progression and a longer life span if fed a fat-restricted diet.

Specifically, Ngo et al. stratified mice between a high-fat diet, in which 42% of the calories came from corn oil, and a low-fat group, in which 12% of calories came from corn oil. The diets were isocaloric. The mice were observed until the tumor progression was noted, upon which these mice were sacrificed. Of note, the mice fed a low-fat diet had a median survival of 20.8 weeks vs. 13.8 weeks for those fed a high-fat diet. Moreover, the mice fed a high-fat diet had a mean prostate specific antigen (PSA) of 75.2 vs. a mean PSA of 10.8 in the mice fed a low-fat diet, suggesting that the high-fat diet resulted in a higher tumor burden [14].

Similar to Ngo et al., other studies have also supported the idea that dietary fat may be linked to prostate tumor progression. Like the Ngo et al. group, mice in the high-fat diet group in other studies were predominantly fed a diet rich in corn oil and in turn had shorter life spans and faster prostate cancer growth rates compared to their low-fat diet counterparts [13,15].

There is no universally agreed upon proposed mechanism that explains the link between high dietary fat intake and prostate tumor growth. However, the results of the aforementioned studies could partly be explained by the fact that high-fat diets utilized corn oil, which is also widely used in western diets and processed foods. Corn oil composition is approximately 60% linoleic acid, an omega-6 fatty acid [38]. Linoleic acid, in turn, is converted into arachidonic acid and prostaglandin E2 (PGE2) via a cyclooxygenase enzyme [39]. In vitro studies have revealed that prostate cancer cells have a 5- to 10-fold higher concentration of arachidonic acid and PGE2 than benign neighboring cells.

In addition, much attention has recently been brought to the importance of the omega-3 to the omega-6 ratio in the diet [40]. In vitro studies by Bagga et al. utilizing fibroblasts have shown that omega-6 fatty acids promote IL-6 production and in turn increase the mitogenic activity of fibroblasts. Notably, this mitogenic activity was attenuated with the introduction of omega-3 or polyunsaturated fatty acids [19]. Cancer-associated fibroblasts make up the stromal component of prostate cancer; increased mitogenic activity of fibroblasts is suggestive of increased cancer aggressiveness [41].

Interestingly, recent literature utilizing the no-carbohydrate ketogenic (NCKD) and low-carbohydrate diets in mice models suggests that such high-fat diets could also be employed to slow prostate cancer progression. Unlike previous high-fat diet models that used corn oil as the primary source of calories, the NCKD and low-carbohydrate models utilized lard and milk fat as the primary source of calories [18]. These studies found that the prostate cancer mice models on the NCKD and low-carbohydrate diets vs. those on the western diet model in which calories were obtained in the following manner: 35% fat, 49% carbohydrates, and 16% protein kcals, had slower tumor growth [16]. In addition, NCKD and low-carbohydrate models also had a lower in vivo ratio of insulin growth factor-1 (IGF-1) to insulin growth factor binding protein-3 (IGFBP-3) [17]. It could be postulated that a decreased ratio of IGF-1:IGFBP-3 could result in anti-cancer properties as IGF-1 has been found to be pro-oncogenic and IGFBP-3 was found to be anti-oncogenic and pro-apoptotic in in vitro prostate cancer models by attenuating angiogenesis and vascular growth [42,43].

Though data from NCKD and low-carbohydrate diet models may seem contradictory of prior high-fat diet models, it should be kept in mind that a high-fat diet via corn oil ingestion vs. a high-fat diet via the ingestion of lard and/or milk fat are different in their micronutrient intake. Corn oil is predominantly rich in omega-6 fatty acids and the ratio of omega-3 to omega-6 is 1:50 [44]. Lard has an omega-3 to omega-6 ratio of approximately 1:10 [45]. As such, a lard diet has approximately five times the intake of omega-3. Such variance in micronutrients could possibly explain the difference in oncogenesis between the two different high-fat diets, as the aforementioned prior in vitro studies by Bagga et al. have revealed that diets with a low omega-3 to omega-6 ratio are pro-inflammatory, and as a result, could be more oncogenic.

Lastly, there is growing interest in the role of cruciferous vegetable intake and prostate cancer outcomes. One major bioactive compound found in cruciferous vegetables is sulforaphane, a compound that is a metabolite of phytochemical compounds known as isothiocyanates [46]. A double-blind, randomized, placebo-controlled multicenter trial with sulforaphane in 78 patients conducted by Cipolla et al. in patients with recurrent prostate cancer showed that individuals administered 60 mg of sulforaphane for six months had on average an 86% longer PSA doubling time [11]. Moreover, a phase II clinical trial with 20 patients with recurrent prostate cancer conducted by Alumkal et al. showed that the administration of 200 μmoles/day of sulforaphane-rich extract resulted in an approximate 50% reduction in PSA doubling time [12]. Though the effect of overall survival was not studied, the decrease in PSA doubling time does suggest that cruciferous vegetable intake could slow prostate tumor progression and thus improve outcomes. However, further studies are needed to determine the amount and ideal preparation of cruciferous vegetables to obtain maximum benefit.

## 5. Malignant Gliomas

Malignant gliomas are aggressive tumors with a poor prognosis. Current treatment modalities include the combination of surgery and chemoradiotherapy. Unfortunately, the complete resection of malignant gliomas has been difficult to achieve as most tumors are noted to recur. This likely occurs because most gliomas extend deep into CNS tissue and are not amenable to wide resection techniques. While chemoradiotherapy has shown promise to extend overall survival, it has its limitations. Whole brain radiation is cytotoxic and can lead to long-term CNS toxicity, which can be problematic for patients who achieve long-term remission from malignant gliomas [47].

Interest in alternative, dietary-based techniques to treat malignant gliomas has been increasing. The basis for this interest is largely founded in studies that showed that although normal brain tissue can utilize both glucose and ketone bodies for metabolism, malignant gliomas are entirely reliant on glucose for energy. It is not completely clear as to why malignant gliomas are unable to utilize ketones; however, it is hypothesized that this could be secondary to underlying mutations affecting the tricarboxylic acid cycle (TCA) cycle and/or the electron transport chain that prevents ketones from entering the TCA cycle as acetyl CoA [48]. The results of such model studies have shown similar physiological changes in patients with malignant gliomas. In patients with high-grade gliomas, tumor tissue has been noted to have a lower levels of glucose and higher levels of lactate than surrounding peri-tumor tissue [49]. This supports the idea that high-grade gliomas are hyper-glycolytic and their metabolism favors lactic acid fermentation via aerobic glycolysis rather than oxidative phosphorylation [50].

The above findings have led to in vivo studies that examine the effects of ketogenic diets (KDs) on malignant gliomas. Malignant glioma mouse models fed a KD with a high fat to carbohydrate weight ratio had slower rates of tumor growth and longer survival than their counterparts, who were fed standard chow (7% simple sugars, 3% fat, 50% polysaccharide, 15% protein as percentage of calories). Placing mice in ketosis also led to a reduction in the level of reactive oxygenation species (ROS) [23,51]. ROS result with the reduction of oxygen and are toxic to normal tissue, and can result in the activation of oncogenes to promote metastases [52]. KDs have also been trialed in conjunction with radiotherapy in mice models. Mice inoculated with malignant glioma cells, treated with radiotherapy, and maintained on a KD, again, with a fat to carbohydrate ratio of 6:1, had significantly longer life spans than their counterparts maintained on standard chow. Such findings suggest that the benefits of a KD are additive to traditional radiotherapy, as the lifespan of mice maintained on a KD and treated with radiotherapy also lived longer than the mice maintained on a KD alone and without radiotherapy [22].

Human studies evaluating the effects of KDs are currently limited. However, several case reports do show that KDs are efficacious in patients with malignant gliomas. A case report of a 65-year-old female with a diagnosis of glioblastoma multiforme treated with concomitant restricted KD with a fat to carbohydrate ratio of 4:1 and chemoradiotherapy showed that such an approach resulted in positron emission tomography (PET)-negative disease. Such resolution in an elderly patient is highly unusual, especially since she did not receive complete surgical resection. Unfortunately, this patient was noted to develop hyperuricemia related to her KD and was subsequently placed on a non-KD diet. Approximately 10 weeks post-transition to a non-KD diet, the patient was noted to develop recurrence on magnetic resonance imaging (MRI). At the time of publication (2010), the patient was alive and being treated with bevacizumab and irinotecan [21]. Similarly, two pediatric patients with astrocytoma tumors treated with KD via a diet composed of 60% fat with medium chain triglyceride oil were also noted to present benefits. Within 7 days of KD, the patients were noted to have 20 to 30 times improvement in ketone levels and were noted to have an average 21% decrease in PET uptake [20].

The above studies and reports show that KDs likely target malignant gliomas by hindering their metabolic needs. Such diets may provide great benefit when used in conjunction with standard therapy.

## 6. Breast Cancer

Breast cancer is the most common invasive cancer and leading cause of cancer-related death in women [53]. The incidence of breast cancer remains lowest in underdeveloped nations. In developed, industrialized nations, the incidence of breast cancer was noted to dramatically increase after 1970. The rise in incidence in part can be explained by the increased utilization of screening mammograms, which ultimately leads to capturing previously undiagnosed breast cancer. However, a major component driving the increased incidence of breast cancer is likely dietary changes from a plant-based, vegetarian diet to a diet more abundant in processed meats and sugar [25]. Recently, data have emerged indicating that a diet rich in vegetables, particularly of the cruciferous types, and fruits, can inhibit the progression of breast cancer by a myriad mechanisms. The following paragraphs will outline in vitro and in vivo experiments that utilized dietary extracts in order to argue for the potential role of dietary intervention in the treatment of breast cancer.

Cruciferous vegetables are from the family *Brassicaceae* and include various species; examples of cruciferous vegetables include kale, broccoli, collard greens, and cabbage. A major bioactive compound in cruciferous vegetables is 3,3′-diindolylmethane (DIM), which is metabolized from precursor compounds known as glucosinolates [54]. DIM is an active compound that has been shown to arrest the proliferation of both estrogen-dependent and -independent cell lines. Though the exact mechanism is not clear, it is suggested that DIM attenuates de novo lipogenesis by inhibiting fatty acid synthase and attenuating levels of the Sp1 transcription factor, which is a major factor that regulates lipogenesis [27]. Moreover, DIM has also been shown to downregulate the expression of urokinase plasminogen activator (UPA), a protein that degrades the ECM and is postulated to play a key role in breast cancer metastasis, and DIM was noted to attenuate vascular endothelial growth factor (VEGF) and MMP-9 levels—proteins that are noted to modulate cell growth and migration. Lastly, DIM was noted to reduced cytokine receptor CXCR4 and CXCL12, which are signaling receptors associated with metastatic growth [26,55]. Additionally, DIM is also noted to inhibit the mammalian target of rapamycin (mTOR), a molecule that is well documented to regulate the cell cycle. Clinically, this is significant as cancers noted to overexpress mTOR are much more likely to suffer recurrence [56].

Unfortunately, the clinical data pertaining to vegetable intake and breast cancer prognosis have been inconclusive. The Women’s Healthy Eating and Living (WHEL) study was implemented to evaluate the effects of a high-vegetable, low-fat diet on breast cancer prognosis and recurrence. Over 2400 women were recruited from 1995 to 2000 and followed until 2006. The women were counseled on increasing vegetable and fruit intake, and plasma carotenoid levels were checked to verify such changes. Interestingly, such changes did not alter breast cancer recurrence rates or confer improvement in all-cause mortality [57]. However, a secondary analysis of the WHEL study revealed that women on tamoxifen, an anti-hormonal drug for ER-positive breast cancer, whose vegetable intake was in the highest tertile, had a significantly lower hazard ratio for recurrence: 0.69, 95% CI 0.55–0.87. Moreover, women whose reported cruciferous vegetable intake was in the highest tertile had the lowest reported hazard ratio of recurrence: 0.48, 95% CI 0.32–0.70 [24]. The latter findings suggest that a high-vegetable diet, particularly rich in cruciferous vegetables, may confer preferential benefit to women undergoing tamoxifen therapy.

Like the WHEL study, the earlier Women’s Intervention Nutrition Study (WINS) also evaluated dietary intervention in early-stage breast cancer patients. In WINS, 2437 women with early-stage breast cancer were enrolled from 1995 to 2001. Of the 2437 women, 975 women were given counseling on a low-fat diet. The remaining patients were not given dietary intervention. The analysis of data from WINS revealed that the low-fat diet group had a lower mean body weight and lower recurrence of breast cancer with a hazard ratio of 0.76, though, with a 95% confidence interval of 0.60–0.98 suggesting that the observed benefit was marginal [58]. Similarly, the analysis of the After Breast Cancer Pooling Project, which included prospective data from breast cancer survivors, showed that cruciferous vegetable intake was not associated with breast cancer outcomes, regardless of the stage, estrogen status, or tamoxifen use [59].

As demonstrated by the above studies, data pertaining to diet and its role in improving breast cancer outcomes are conflicting. Pre-clinical data from in vitro and in vivo studies suggest that bioactive compounds like DIM may have a potential role in modulating breast cancer biology. However, clinical data are equivocal as the WHEL study and WINS did not display clear benefit from increasing the vegetable intake, though a subgroup analysis of the WHEL study did suggest that cruciferous vegetable intake may improve outcomes in ER-positive patients on tamoxifen. Further clinical investigation is warranted to further assess the role of vegetable intake in improving breast cancer outcomes.

## 7. Immunotherapy

Immunotherapy is an emerging modality in the treatment of cancer. For the better part of the last five decades, the hallmark of cancer treatment has been chemotherapy and radiotherapy; these modalities were employed due to their cytotoxic effects. Unfortunately, both chemotherapy and radiotherapy lack specificity and can result in the damage of healthy tissue in addition to targeting cancerous tissue. However, in the last decade, immunotherapy has become an emerging modality in oncology. At its core, immunotherapy is based on the theory that the immune system has an inherent mechanism to prevent the onset of cancer via immune surveillance from NK cells, CD4+, and CD8+ T cells. Such surveillance targets aberrant cells and prevents the onset of clinical malignancy [60].

Immunotherapy has provided a paradigm shift in how to treat cancer. It has become the first-line treatment in many cancer states, including lung, melanoma, genitourinary, and rare disease states that have a high tumor mutational burden. We feel that such a novel approach, which manipulates the immune system to fight cancer, is likely more responsive to dietary changes to improve outcomes than traditional chemotherapy as recent studies have shown that the gut microbiome plays a crucial role in determining the overall response rate to such therapy [61].

Key components of immune surveillance are the actions of cytotoxic T-lymphocyte-associated protein-4 (CTLA4) and programmed death-1 (PD-1) protein. These proteins are critical in regulating T cell response. CTLA4 is an inhibitory protein expressed by T cells that competes with the CD28 marker on T cells for interaction with B7-1 and B7-2 on antigen presenting cells (APCs). CD28 interaction with B7-1/B7-2 is required for the activation of T cells, and thus, when CTLA4 interacts with B7-1/B7-2 instead of CD28, it prevents the activation of T cells and leads to the downregulation of the immune response. Similarly, PD-1 is expressed by T cells and if it interacts with the programmed death-1 ligand (PD-L) on APCs, it leads to the attenuation of the intracellular signaling cascade initiated by CD28 priming and thus also results in the downregulation of the immune response [62].

Clinical and basic science studies have shown that certain cancers can utilize CTLA4 and PD-1/PD-L to evade immune surveillance. As a result, several monoclonal antibodies targeted against CTLA4 or PD-1/PD-L have gained FDA approval. These monoclonal antibodies are commonly referred to as checkpoint inhibitors (CPIs). The mechanism of CPIs is to prevent the immune system’s tolerance of cancer cells. The blockade of CTLA4 to B7-1/B7-2 and PD-1 to PD-L results in an inflammatory response and the recognition and destruction of cancerous cells [63].

However, the response rate to CPIs is not uniform. CPIs have a response rate of approximately 10–30%. It remains unclear as to why the response rate is so diverse, especially since CPIs are often administered in tumors that have documented the expression of PD-L where, in theory, blocking the interaction of this inhibitory protein should elicit an inflammatory response and result in therapeutic benefit. Recently, it has emerged that the gut microbiome can a modulate response to both immunotherapy and chemotherapy. Iida et al. evaluated the effects of change in gut flora to chemotherapy response in mice. The group utilized a murine cancer model where EL4 lymphoma, MC38 colon carcinoma, and B16 melanoma cells were injected subcutaneously. The mice were divided into two groups: one group was treated with antibiotics prior to the initiation of anti-IL-10 antibody treatment, which is well documented to cause hemorrhagic necrosis in cancer models via upregulating the tumor necrosis factor (TNF), and another group of mice was not treated with antibiotics. In this study, the group of mice pretreated with antibiotics did not respond to anti-IL-10 antibody treatment and had a significant progression of disease. Mice not treated with antibiotics had a significant response to the anti-IL-10 antibody and had a significantly longer life span. Prior antibiotic treatment was also noted to attenuate TNF levels and cause gut dysbiosis.

As expected, antibiotic treatment decreased the gut flora and diversity, which both recovered after antibiotic cessation. Interestingly, the Gram-negative genera of *Alistipes* and Gram-positive genera of *Ruminococcus* were noted to be positively correlated with TNF levels, response to therapy and with circulating levels of NK cells and macrophages. Moreover, in mice previously pretreated with antibiotics, the transplantation of *Alistipes* species improved the response to therapy and improved survival. These findings suggest that the gut flora could very well regulate locoregional inflammatory markers and regulate response to immunotherapy [30]. 

The gut flora of patients undergoing immunotherapy and its significance in modulating response has also been evaluated. Gopalakrishnan et al. prospectively examined the gut flora of patients undergoing immunotherapy for metastatic melanoma to decipher if there was a difference in the flora between responders and non-responders to immunotherapy. Interestingly, the group found that patients who responded to immunotherapy were enriched with bacteria from family *Ruminococcaceae*, especially from the genus *Faecalibacterium*, and non-responders were enriched with bacteria from family *Bacteroidales*. Furthermore, within the responder group, patients who had a higher abundance of *Faecalibacterium* had a significantly longer progression free survival (PFS) than those who had a lower abundance. On the contrary, within the non-responder group, individuals who had a higher abundance of *Bacteroidales* had a significantly lower PFS than those who had a lower abundance. The mechanism behind such difference in response to immunotherapy between *Faecalibacterium*-predominant patients and *Bacteroidales*-predominant patients is likely secondary to the immunomodulatory effects of these bacteria. Responders with a high abundance of *Faecalibacterium* had a higher circulating load of CD8+ T cells and antigen presenting cells (APCs) in circulation and in the local tumor environment than those who had a higher abundance of *Bacteroidales* [29].

More importantly, a follow-up abstract revealed that patients who were responders to immunotherapy and who had a higher abundance of bacteria from genera *Ruminococcaceae* had a diet rich in fiber and rich in fruits and vegetables. Conversely, non-responders were reported to have a diet rich in processed sugars and processed meats. It is argued that a high-fiber diet promotes a diverse microbiota and a gut flora rich in *Ruminococcaceae* species [28]. This favorable flora likely promotes higher levels of circulating CD8+ T cells and APCs, which is critical to the efficacy of CPIs. CPIs are dependent on an effective immune response and the cytotoxic effects of activated CD8+ T cells. Bacteria from genera *Ruminococcaceae* and *Faecalibacterium*, are correlated with increased APC and CD8+ T cell levels, which can explain why they are associated with CPI response—they arguably increase antigen presentation and immune surveillance, and ultimately, increase the efficacy of CPIs.

## 8. Discussion

Interest continues to rise in the potential role of dietary intervention to improve cancer prognosis. However, at this point, the majority of the data are pre-clinical and stems from in vitro and in vivo animal studies. Prospective studies and meta-analyses like the WHEL study and WINS, which showed marginal benefit in the dietary intervention group, had inherent limitations as they did not control for confounding factors nor did they standardize a diet. However, recent publications showcasing the important role of the gut microbiome in predicting outcomes in immunotherapy and colon cancer give us insight into how to arrange future studies.

Fiber intake increases SCFAs in the gut and diversifies flora to what is believed to be most beneficial to promote anti-neoplastic properties. Increased SCFAs in the gut on CRC rat models led to longer life spans and less tumor invasion. In addition, in in vitro models, SCFAs have shown to decrease the levels of pro-inflammatory cytokines and proteins like MMP-9, which are well documented to promote tumor invasion [64]. Furthermore, the patients undergoing immunotherapy who were noted to have a diversified gut microbiome with predilection to species from genera *Ruminococcus*, were noted to more likely respond to treatment than those with less diversified flora. Future studies should examine the effects of a high-fiber diet on CRC patients and those receiving immunotherapy, as fiber intake has been shown to promote gut flora that is both rich in diversity and rich in species from the phylum *Firmicutes* [65].

For prostate and malignant gliomas, high omega-3 intake and ketogenic diets have shown to provide clinical benefit, respectively. Historically, high-fat intake was seen as a risk factor for prostate cancer. However, such studies were mostly animal models that utilized corn or other processed oils to increase fat intake. When the fats were changed to foods with a more balanced omega-3 to omega-6 ratio, animal studies showed improved outcomes. Malignant gliomas have also been noted to have improved outcomes with a high-fat, ketogenic diet. Initially, this benefit was believed to be in part due to the Warburg effect, which argues that cancer relies on glycolysis for metabolism, and by eliminating carbohydrates from the diet it “starves” the cancer of its nutrients. Though there may be conflicting data regarding the clinical benefit of ketogenic diets in malignant gliomas, it is essential to examine the ratio of fats to carbs in the diet. Experiments that have failed to show the benefit of ketosis have utilized diets with a fat to carbohydrate ratio of approximately 3:1 [66]. On the contrary, experiments that have shown clinical benefit from ketogenic diets have mostly utilized diets with a fat to carbohydrate ratio of 6:1.

In summary, the current data regarding diet and cancer prognosis are limited. Most to date is being extrapolated from pre-clinical studies to human subjects and makes broad generalizations across all cancer types. There is a great need for further studies in order to make evidence-based recommendations for dietary changes to cancer patients as diet is an integral part of cancer care. Specifically, further studies are needed, preferably in a case-controlled manner, where patients are provided intervention (e.g., high-fiber diet in CRC and patients receiving immunotherapy) and biomarkers (e.g., changes in microbiota) are compared to those who did not receive intervention and served as controls. Such studies will allow us to better understand the effects of diet on the human body, and allow us to make more meaningful and evidence-based recommendations in order to improve survivorship and prognosis.

## Figures and Tables

**Table 1 nutrients-12-02226-t001:** Summary of Studies Reviewed to Improve Outcomes in Specific Cancer Patients.

	Study	Design	Primary endpoint	Results	Dietary Recommendations
**Colon cancer**	O’Keefe et al. [4]	Clinical; 40 patients	Effect of high-fiber diet on microbiota	High-fiber diet promoted diversification of gut flora, decreased Ki67 expression, and decreased macrophage and lymphocyte presence in colonic tissue	1. Plant-based diet with fiber intake approaching 50g/d
	Segain et al. [5]	Clinical; 17 patients	Effect of butyrate enemas on colonic inflammation	SCFAs decreased TNF production and pro-inflammatory cytokine mRNA expression	
	Borges-Canha et al. [6]	Clinical; systematic review of 31 studies	Link between microbiota and colon cancer	Microbiota dysbiosis was suggestive of colorectal carcinogenesis	
	Van Blarigan et al. [7]	Clinical; prospective cohort; 992 patients	Effect of fruit and vegetable diet, healthy body weight, and increased physical activity on survival in stage 3 colon cancer	Plant-based diet, physical activity, and healthy body weight was associated with longer survival	
	Soret et al. [8]	Preclinical; rat model	Effect of SCFAs on enteric nervous system	SCFAs increased cholinergic-mediated muscle contractile response	
	Yue et al. [9]	Preclinical; mouse model	Effect of *Lactobacillus plantarum* on colon cancer progression and locoregional inflammation	*L. plantarum* inhibited tumor development and locoregional inflammation	
	Escamilla et al. [10]	Preclinical; in vitro	Effect of *Lactobacillus* spp. on colon cancer invasion	*Lactobacillus* supernatants inhibited metastatic ability	
**Prostate cancer**	Cipolla et al. [11]	Clinical; double-blind randomized controlled trial; 78 patients	Effect of sulforaphane intake on PSA doubling time	Intake of sulforaphane was associated with 86% longer doubling time	1. Plant-based diet with high cruciferous vegetable intake, particularly vegetables containing sulforaphanes 2. Low-carbohydrate, ketogenic diet with high omega-3 intake
	Alumkal et al. [12]	Clinical; phase 2; 20 patients	Effect of sulforaphane intake on PSA doubling time	Intake of sulforaphane was associated with approximately 50% longer doubling time	
	Aronson et al. [13]	Preclinical; mouse model	Effect of fat intake on prostate cancer tumor progression	Fat-restricted diet slowed tumor progression	
	Ngo et al. [14]	Preclinical; mouse model	Effect of fat intake on prostate cancer tumor progression	Fat-restricted diet slowed tumor progression	
	Wang et al. [15]	Preclinical; mouse model	Effect of fat intake on prostate cancer tumor progression	Fat-restricted diet slowed tumor progression	
	Caso et al. [16]	Preclinical; mouse model	Effect of carbohydrate restriction on prostate cancer tumor progression and insulin axis	Carhobhydrate restriction slowed tumor progression and decreased insulin levels; ratio of IGF to IGFBP lowered however not statistically significant	
	Freedland et al. [17]	Preclinical; mouse model	Effect of NCKD on prostate cancer tumor progression and insulin axis	NCKD decreased tumor progression compared to western diet, lowered insulin and IGF levels, and increased expression of IGFBP	
	Masko et al. [18]	Preclinical; mouse model	Effect of carbohydrate restriction on prostate tumor progression and insulin axis	Carhobhydrate restriction slowed tumor progression and decreased insulin levels; ratio of IGF to IGFBP lowered however not statistically significant	
	Bagga et al. [19]	Preclinical; in vitro	Effect of omega-6 on production of inflammatory cytokines	Excess omega-6 promoted increased IL-6 production and mitogenic activity of fibroblasts; omega-3 attenuated this response	
**Malignant gliomas**	Nebeling et al. [20]	Clinical; case report; 2 patients	--	A ketosis diet resulted in 21% decrease in PET-avidity; 1 patient maintained on ketosis diet for 12 months and did not experience disease progression	1. Ketogenic diet
	Zuccoli et al. [21]	Clinical; case report; 1 patient	--	Ketogenic diet with fat to carbohydrate ratio of 4:1 (as percentage of calories) resulted in PET-negative disease	
	Abdelwahab et al. [22]	Preclinical; mouse model	Effect of ketogenic diet on radiation therapy for malignant glioma	Ketogenic diet enhanced anti-tumor effects of radiation	
	Stafford et al. [23]	Preclinical; mouse model	Effect of ketogenic diet on malignant glioma progression	Ketogenic diet slowed tumor progression and decreased reactive oxygen species production	
**Breast cancer**	WHEL study	Clinical; randomized controlled trial; 2,448 patients	Role of dietary pattern in prognosis	No statistical difference in breast cancer recurrence, improvement in prognosis, or all-cause mortality with adherence to plant-based diet	1. Plant-based diet with high cruciferous vegetable intake, particularly vegetables containing sulforaphanes
	Thomson et al. [24]	Clinical; sub-group analysis of WHEL study	--	Women with hormone receptor-positive breast cancer on tamoxifen who adhere to plant-based diet with high cruciferous vegetable intake may have benefit in breast cancer recurrence	
	WINS study	Clinical; randomized controlled trial; 2,437 patients	Effect of low-fat diet on early stage breast cancer	Adhering to a low-fat diet post-treatment resulted in lower recurrence rates	
	After Breast Cancer Pooling Project	Clinical; prospective cohorts; 18,314 patients (84% stage 1-2 breast cancer)	Effect of physical activity, dietary factors, and quality of life in breast cancer prognosis	Vegetable intake was not associated with breast cancer outcomes	
	Ghoncheh et al. [25]	Clinical; retrospective case-control	Risk factors for breast cancer	Diet rich in processed meats and refined carbohydrates was a risk factor for breast cancer	
	Thomson et al. [26]	Preclinical; review article; animal model and in vitro	Effect of DIM, a major bioactive compound in cruciferous vegetables, on breast cancer growth	DIM inhibited breast cancer tumor growth by downregulating UPA, which controls VEGF and MMP-9 production; DIM reduced cytokine receptor CXCR4 and CXCL12, which are signaling receptors associated with metastatic growth	
	Saati et al. [27]	Preclinical; in vitro	Effect of DIM on breast cancer cell growth	DIM inhibited breast cancer line growth likely by inhibiting expression of transcription factor Sp1	
**Immunotherapy**	Spencer et al. [28]	Clinical; prospective cohort; 113 patients	Relationship between lifestyle factors and response in melanoma patients undergoing immunotherapy	Patients with high-fiber diet noted to have highest odds of response to immunotherapy	1. High-fiber diet
	Gopalakrishnan et al. [29]	Clinical; prospective cohort; 112 patients	Relationship between gut microbiome and response to immunotherapy in metastatic melanoma patients	Patients with abundance of *Clostridiales* were noted to have higher response to immunotherapy, while non-responders were predominantly rich in *Bacteroidales*	
	Iida et al. [30]	Preclinical; mouse model	Effect of modulating tumor micro-environment on IL-10 immunotherapy response	Antibiotic treatment induced gut microbiome changes, attenuated IL-10 response	

Abbreviations: CXCL12: C-X-C motif chemokine 12, CXCR4: C-X-C motif chemokine receptor 4, DIM: 3,3′-diindolylmethane, IGF: insulin growth factor, IGFBP: insulin growth factor binding protein, IL-6: interleukin-6, IL-10: interleukin 10, MMP-9: matrix metalloproteinase-9, NCKD: no-carbohydrate ketogenic diet, PET: positron emission tomography, PSA: prostate specific antigen, SCFAs: short chain fatty acids, Sp1: specificity protein 1, TNF: tumor necrosis factor, UPA: urokinase plasminogen activator, VEGF: vascular endothelial growth factor.

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
