# Peer review of "Understanding the Mechanisms of Diet and Outcomes in Colon, Prostate, and Breast Cancer; Malignant Gliomas; and Cancer Patients on Immunotherapy"

_nutrients, 2020, doi:10.3390/nu12082226_

Round 1

Reviewer 1 Report

In this review, authors provided an overview of existing studies that investigated the role of different diets on cancer progression. This is an interesting review that is addressing topic which would attract broader scientific community. It is very well written and understandable.

There is another dietary program that apparently showed effect on longevity and cancer incidence and that is Calorie Restriction diet. How CR was found to affect cancer progression in any of the analyzed cancer types and if authors can address that in their review since it is another diet plan that is becoming popular?

Reviewer 2 Report

This review handles the problem of dietary interventions in different types of cancer quite well. I have some comments:

  • I would propose that the different types of cancer and their respective associated diets, are summarized in a table.
  • The "Immunotherapy" section, while interesting as such, does not seem to really "fit" into this article, in my personal view.
  • A figure could be added to show the effects of dietary interventions on cancer patients receiving immunotherapy.
